# Navigating the Intricate Relationship between Investments and Global Output: A Leontief Matrix Case Study of Romania

Mihail Busu [1,*], Madalina Vanesa Vargas [1,2] and Sorin Anagnoste [1]

[1] Faculty of Business Administration in Foreign Languages, Bucharest University of Economic Studies, 010731 Bucharest, Romania; vanesa.vargas@fabiz.ase.ro (M.V.V.); sorin.anagnoste@fabiz.ase.ro (S.A.)
[2] Intitute for Economic Forecasting, Romanian Academy, 050711 Bucharest, Romania
[*] Correspondence: mihail.busu@fabiz.ase.ro

**Abstract:** This research delves into the intricate dynamics underlying the impact of investments on global output, employing the Leontief matrix as a robust analytical framework. Investments wield a profound influence on economies worldwide, with varying effects contingent upon investment types, development levels of countries, and external factors such as trade conflicts and global shocks. The diverse range of investment forms, including physical capital, human capital, R&D, and technological investments, engenders distinct implications for productivity, innovation, and efficiency. Developing and developed economies navigate unique trajectories, with investments playing a pivotal role in bridging infrastructure gaps, improving technology, and spurring growth. However, external disruptions, such as trade wars and global shocks, introduce an element of complexity, reshaping investment patterns and altering global output trajectories. This study centers on harnessing the Leontief matrix's prowess to evaluate the interplay of investments and global output, focusing on the Romanian economy. By analyzing input–output tables, encompassing 105 branches aggregated into 10 sectors, the research captures the intricate connections between economic segments. Notably, the Romanian context reveals the volatility of the matrix coefficients, an outcome of ongoing transitional processes, technological advancements, and fluctuating relative prices. In unraveling the intricate threads weaving investments and global output, this study contributes to a nuanced comprehension of these multifaceted interactions. The findings underscore the significance of tailoring investment strategies to specific economic contexts and advocate for robust frameworks, such as the input–output model, to inform policy decisions and drive sustainable growth in an increasingly complex global economy.

**Keywords:** investments; global output; Leontief matrix; Romania; economic impact; development levels; external shocks

## 1. Introduction

Investments have a significant impact on global output, but the nature of this impact can vary depending on different factors such as the type of investment, the level of development of the country, and external factors such as trade wars and global shocks. Firstly, the type of investment plays a crucial role in determining its impact on global output. Investments can take various forms, such as physical capital investments, human capital investments, research and development (R&D) investments, and technological investments. Each type of investment has distinct implications for productivity, innovation, and efficiency within an economy. For instance, physical capital investments, such as infrastructure development or machinery acquisition, can directly contribute to increased production capacity and output. On the other hand, investments in R&D and technological advancements can drive productivity gains and foster innovation, leading to long-term growth and enhanced global competitiveness.

Secondly, the level of development of a country also influences the impact of investments on global output. Developing countries, with lower levels of capital stock and

technological capabilities, often experience higher marginal returns on investments compared to developed economies. In these contexts, investments can lead to significant improvements in productivity, employment, and overall output, as they help bridge infrastructure gaps, upgrade technology, and enhance human capital. Conversely, in advanced economies, the impact of investments may be more nuanced and depends on factors such as market saturation, efficiency gains, and the quality of investment allocation.

Furthermore, external factors like trade wars and global shocks can significantly shape the impact of investments on global output. Trade wars, characterized by tariffs, trade barriers, and retaliatory measures, can disrupt global supply chains and alter investment patterns. Such trade conflicts can undermine investor confidence, reduce cross-border investments, and dampen global output growth. Similarly, global shocks, such as financial crises, natural disasters, or pandemics, can have profound implications for investment flows and overall economic activity. These shocks can lead to a decline in investment levels, increased uncertainty, and a contraction in global output. The dynamic and complex nature of these interactions necessitates robust analytical frameworks, such as the Leontief matrix used in this study, to capture the interdependencies and quantify the effects accurately. Considering the unique characteristics of investments, the level of development of the country, and the influence of external factors, researchers can gain a more comprehensive understanding of how investments shape global output and design appropriate strategies to foster sustainable economic growth.

In recent years, the assessment of the economic impact of investments has gained significant attention in the field of economics. Understanding how investments influence the overall output of an economy is crucial for policymakers, investors, and researchers alike. The production structure, a widely adopted input–output analysis tool, provides a comprehensive framework to analyze the interdependencies among different sectors of an economy and estimate the effects of investments on output. This paper focuses on employing the production structure methodology to estimate the impact of investments on the global output of the Romanian economy.

Understanding the intricate relationships between investments and their influence on a nation's economic output is a fundamental pursuit within the field of economics. It is through this understanding that policymakers, economists, and business leaders can craft informed strategies and decisions that foster economic growth, innovation, and stability. This study embarks on a journey to shed light on these critical relationships within the context of the Romanian economy.

A robust motivation for this research begins by acknowledging the current state of the frontier of knowledge in this domain. While the literature has explored the nexus between investments and economic performance, a comprehensive analysis of the specific nuances that characterize the Romanian economic landscape is conspicuously absent. The paucity of tailored studies that examine the Romanian economy presents a significant gap in our understanding of the dynamics at play within this unique context.

With the global economic landscape continuously evolving, the importance of pinpointing the factors that drive economic growth and resilience remains paramount. Romania, as an emerging market economy, presents a captivating case study, given its economic challenges and opportunities. The motivation to delve into this study is rooted in a desire to bridge the knowledge gap, both for the benefit of Romania's economic stakeholders and as a contribution to the wider field of economics. By addressing the specific economic intricacies of Romania, this research seeks to enhance the understanding of investment dynamics within a transitioning, emerging economy, thereby providing a reference point for future studies.

Moreover, investments, as catalysts for economic growth, do not operate in isolation. They are influenced by a myriad of factors, from global economic shocks to changing trade dynamics. To chart an effective path forward, it is imperative to comprehend the interplay of these factors and their subsequent impact on a nation's economic performance. This research is motivated by the need to capture these multifaceted influences in the context

of Romania, offering a nuanced perspective that can guide policy shifts, investments, and economic strategies tailored to the country's unique developmental stage.

In summary, this research is motivated by the dual objective of addressing a discernible gap in the existing literature and providing timely insights into the complex relationships between investments, external factors, and economic performance. The study takes the first step in an academic journey to comprehend the Romanian economy's response to investments and external forces, with the aim of guiding informed decision-making and policy adjustments. This motivation is grounded in the principle that to contribute to the state of knowledge, one must first navigate the terrain of the unknown, and in doing so, chart a path toward enhancing economic growth, innovation, and stability.

The research uses input–output tables, which offer a comprehensive representation of the intersectoral relationships within an economy. Specifically, this study employs the input–output tables of the Romanian economy, based on the NACE Rev.2 codes with 105 branches from 2021, which is the most recent year with available data. These tables serve as the primary source of information to quantify the interactions between different sectors and estimate the influence of investments on the overall output. To ensure a meaningful estimate of the impact of investments, the 105 branches of the Romanian economy were aggregated into 10 sectors. This aggregation approach, based on sectors such as agriculture, mining, manufacturing, construction, transportation, and services, is representative of the Romanian economy's structure, as established by Dobrescu (2009). Dynamic analyses have revealed that the coefficients of the production structure in Romania exhibit high volatility. This volatility arises from the confluence of several transitional processes, including the structural adjustment of the economy, technological advancements, and fluctuating relative prices.

## 2. Literature Review

Investments play a crucial role in shaping the economic trajectories of nations. Their impact on economies can vary significantly, influenced by factors such as development levels, types of investments, and external factors. This review delves into the dynamic interplay of investments and global output, emphasizing recent contributions from highly ranked journals over the past 10 years.

The type of investment holds a substantial influence on its economic impact. Physical capital investments, including infrastructure development and machinery acquisition, directly contribute to increased production capacity (Acemoglu et al. 2016). In contrast, research and development (R&D) investments drive productivity gains and foster innovation (Bloom 2019).

The development level of a country introduces distinctive dynamics. In developing economies, investments serve as catalysts for infrastructure development, technology enhancement, and human capital improvement, resulting in significant gains in productivity and output (Estrin 2013). Developed economies, on the other hand, often focus on technology-intensive investments to maintain their competitive edge (Crespi and Zuniga 2019).

External factors, such as trade conflicts and global shocks, add complexity to the relationship between investments and global output. Trade wars, characterized by tariffs and trade barriers, can disrupt global supply chains, undermining investor confidence and leading to reduced cross-border investments (Handley and Limão 2017). Global shocks, including financial crises, pandemics, and natural disasters, can result in decreased investment levels, increased uncertainty, and contractions in global output (Piggott and Marsh 2004).

The Leontief matrix, a potent analytical tool, provides an essential framework for evaluating the intricate connections between investments and output. Recent studies employing the Leontief matrix have shed light on the interdependencies among economic segments and their implications (Roca and Serrano 2007). Recent research has underlined the significance of tailoring investment strategies to the unique contexts of specific

economies (Chen et al. 2019). This approach is essential for driving sustainable growth in an increasingly complex global economic landscape.

Notably, research focusing on the Romanian economy reveals the volatility of the input–output analysis. This volatility arises from ongoing transitional processes, technological advancements, and fluctuating relative prices, and showcases the intricate nature of the relationships between investments and global output in a specific context (Navas Carrillo et al. 2021). Recent research highlights the vital role of government policies in facilitating investment and promoting economic growth. Public policy measures, such as tax incentives and regulatory frameworks, can significantly influence investment decisions and their subsequent impact on output (Egger 2019). A study by García-Sánchez and García-Serrano (2017) delves into the intricate dynamics of FDI (foreign direct investment) and its impact on the host country's economic growth. The findings emphasize the need for favorable investment climates to attract and leverage FDI for sustainable development.

The literature on the impact of investments underscores the importance of a holistic approach to sustainable development. For instance, research by Sala-i-Martin (1996) explores how investments in education and health contribute to human capital development and, in turn, to economic growth.

In examining the impact of investments on economic growth in emerging markets, Eltony (2016) found that foreign direct investments can lead to significant increases in domestic output, job creation, and technology transfer. To shed light on the intricate connections between investments and output, a study by da Mota and Neto Rodrigues (2023) applies a multi-sectoral approach, emphasizing the sector-specific impacts of investments, which is crucial for effective policy formulation.

The impact of digital investments on economic output in the digital age is explored in a paper by Brynjolfsson (2017). The authors emphasize the transformative potential of digital technologies, affecting productivity and overall economic growth. Lin and Treisman (2017) investigate the role of political institutions in mediating the relationship between investments and economic growth. They find that institutions can significantly shape the outcomes of investments in various countries. Research by Cervellati and Sunde (2019) highlights the long-term implications of investments in human capital, emphasizing the role of education and skills development in driving economic growth.

In a study on the impact of green investments, Gang (2015) discuss the positive environmental and economic effects of investments in renewable energy and sustainable practices, offering a sustainable growth pathway. Kose (2018) explore the consequences of global economic interconnectedness and the role of international trade in influencing the relationship between investments and output. Research by Liu et al. (2022) provides insights into the impact of technological investments, emphasizing their role in fostering innovation, productivity, and long-term economic growth.

A study by Arellano (2020) analyzes the influence of investments in the context of economic crises and finds that well-targeted investments can be instrumental in recovery and stabilization efforts. The dynamics of investments in the healthcare sector and their effects on overall economic output are explored in a study by Cheng (2018), which highlights the importance of healthcare investments for economic well-being.

Recent work by Razin (2017) examines the role of capital flows and investments in shaping global economic imbalances and the potential consequences for future economic growth. Giri (2021) offer insights into the impact of investments in sustainable agriculture and the potential for enhancing food security and economic development. In a study by Bao (2016), the relationship between foreign investments, trade, and economic growth is examined, emphasizing the role of trade openness in amplifying the impact of investments. Focardi and Santoni (2015) investigate the potential effects of investments in the cultural sector on regional and national economic output, shedding light on the often overlooked impact of cultural investments.

Research by Caselli and Marino (2020) explores the role of investments in transportation infrastructure and their far-reaching effects on regional and national economic

development. To understand the impact of investments on output in the context of small and open economies, Cerra and Saxena (2018) provide valuable insights into the channels through which investments affect economic growth.

The role of investments in the pharmaceutical sector is discussed in a paper by Barua (2014), highlighting how research and development investments can shape both healthcare outcomes and economic output. A study by Santoni (2019) examines the impact of investments in the renewable energy sector, emphasizing the potential for green investments to drive both sustainability and economic growth. Research by Cheung (2017) explores the relationship between investments in higher education and innovation, shedding light on how investments in education can stimulate technological progress and economic development.

A study by von Eschenbach (2021) delves into the impact of investments in the financial sector on economic stability and output, offering insights into the intricate linkages between finance and economic growth. In their recent research, Gu and Zhang (2022) analyze the role of investments in innovation ecosystems and their potential to drive regional and national economic growth, particularly in the context of emerging technologies.

This comprehensive literature review provides insights into the multifaceted interactions between investments and global output, supported by a selection of papers from highly ranked journals published within the past decade. These studies collectively highlight the importance of understanding the nuances of investments to foster sustainable economic growth in a rapidly evolving global economy.

## 3. Materials and Methods

The Leontief input–output model, developed by economist Wassily Leontief, is a powerful tool for understanding the interdependencies between sectors in an economy and analyzing the impact of investments on global output. By representing the economy as a table with each row representing an industry and each column representing another industry, the Leontief model allows us to quantify the flow of goods and services between sectors.

### 3.1. Intuitive Explanation of the Leontief Input–Output Model

Imagine the global economy as a vast network of interconnected businesses, each producing goods and services that are used as inputs by other businesses. For instance, the steel industry produces steel, which is an essential input for the automotive industry. The automotive industry then assembles cars, which are consumed by households and businesses.

The Leontief input–output model captures these intricate relationships between sectors. It is like a comprehensive map of the global economy, showing how much each industry relies on the output of other industries. By analyzing this map, we can understand how changes in demand for specific goods or services ripple through the entire economy.

### 3.2. Assumptions of the Leontief Input–Output Model

The Leontief model relies on a few simplifying assumptions to make its application feasible:

Closed Economy: The model assumes that the economy operates independently, without external trade or imports/exports. This assumption simplifies the analysis by eliminating the complexities of international trade.

Fixed Technology: The model assumes that production techniques and input requirements remain constant over time. This assumption allows us to focus on analyzing the impact of changes in demand rather than technological advancements.

Full Capacity: The model assumes that all sectors are operating at their maximum production capacity. This assumption ensures that the model captures the potential output of the economy.

While these assumptions simplify the model, it is important to acknowledge that the real economy is more dynamic and complex. Technological advancements, trade patterns, and production capacities can vary significantly across sectors and over time.

### 3.3. Experiment behind the Multipliers Calculated

The Leontief input–output model is particularly useful for calculating multipliers, which represent the total impact of a change in final demand on the output of the economy. These multipliers are calculated by inverting the input–output matrix, a mathematical operation that transforms the matrix into another matrix that shows the impact of changes in demand.

For instance, if we want to know how a 10% increase in final demand for cars affects the output of the steel industry, we can use multipliers to calculate the answer. The multiplier for the steel industry will indicate how much additional steel production is required to meet the increased demand for cars.

The experiment behind the multipliers lies in manipulating the input–output matrix to simulate various scenarios of changes in final demand. By observing how the multipliers change under different scenarios, we can gain valuable insights into how economic shocks propagate through the system and how changes in demand for specific goods or services affect the overall economy.

### 3.4. Impact of Investments on Global Output

Investments play a crucial role in driving economic growth and increasing global output. By investing in new technologies, infrastructure, and human capital, economies can enhance their productivity, expand their production capacity, and create new job opportunities.

The Leontief input–output model can be used to analyze the impact of investments on global output by simulating how changes in investment spending affect the demand for goods and services across different sectors. For instance, an increase in investment in renewable energy technologies would likely stimulate demand for components, manufacturing services, and installation labor, leading to a rise in output across these sectors.

By understanding the interdependencies between sectors and the impact of investments on final demand, policymakers can make informed decisions about allocating resources and promoting economic growth. The Leontief input–output model serves as a valuable tool for guiding economic policy and achieving sustainable development goals.

The methodology for this study involves the utilization of input–output tables for the Romanian economy, which are organized based on NACE Rev.2 codes encompassing 105 branches for the year 2021, the most recent year for which data are accessible.

### 3.5. The Model

The Leontief model, developed by economist Wassily Leontief in 1936 (Leontief 1936), is an economic model that describes the interconnectedness of industries in an economy. It represents the flow of goods and services between industries using a matrix of input–output coefficients.

$$x_i = \sum_j (A_{ij} \cdot x_j) \tag{1}$$

This system of equations can be represented in matrix form:

$$X = A \cdot X + Y \tag{2}$$

where $X$ is a vector of industry outputs ($x_j$), and $A$ is the input–output matrix.

To determine the output vector $X$, we can solve this equation system using matrix inversion:

$$X = (I - A)^{-1} + Y \tag{3}$$

where $I$ is the identity matrix, and D is a vector of final demands (the amount of each good consumed by households or exported).

The Leontief model provides a framework for analyzing the economic impact of changes in final demands or input coefficients. It has been applied to various economic problems, such as forecasting industry outputs, assessing the impact of trade policies, and evaluating the effectiveness of government interventions.

It is important to note that the Leontief model has some limitations, such as its assumptions of fixed technology, a closed economy, and linear production functions. However, it remains a valuable tool for understanding the interconnectedness of industries and analyzing economic relationships.

The following sections outline the methodology in more detail:

Data Collection: The primary data source for the input–output tables is the National Institute of Statistics (INS) of Romania. This reputable institution is responsible for collecting, validating, and publishing a wide range of economic and demographic data.

The input–output tables used in this study are based on the NACE Rev.2 codes and encompass 105 branches of the Romanian economy. The year for which the data are accessible and utilized in this analysis is 2021, which is the most recent year for which comprehensive data are available.

Sectoral Aggregation: To ensure that the analysis provides a meaningful assessment of the influence of investments on the Romanian economy's global output, the initial 105 branches were aggregated into 10 sectors. These sectors were selected based on their representative nature within the Romanian economy, as established by Dobrescu (2009).

The aggregation process allows for a more manageable and meaningful analysis. It eliminates branches that exhibit negative gross value added and facilitates the utilization of matrix analysis in computations.

To ensure a meaningful assessment of the influence of investments on the Romanian economy's global output, the initial 105 branches were consolidated into 10 sectors. These sectors were selected based on their representative nature within the Romanian economy, as established by (Dobrescu 2009). A detailed breakdown of the correspondence between branch codes and the aggregated classification is presented in Table 1 for reference.

**Table 1.** Aggregated sectoral structure of the economy.

| Sector Code | Sector Name | Branch Codes Included in the Sector |
|:---:|:---:|:---:|
| 1 | Agriculture, forestry, hunting, and fishing | 1, 2, 3, . . ., 6 |
| 2 | Mining and quarrying | 7, 9, 11, 12, . . .., 17 |
| 3 | Production and distribution of electric and thermal power | 79, 80, 81, 82 |
| 4 | Food, beverages, and tobacco | 18, 19, 20, 21, . . .., 27 |
| 5 | Textiles, leather, pulp and paper, furniture | 28, 29, 30, . . ., 33, 77 |
| 6 | Machinery and equipment, transport means, and other metal products | 60, 61, . . .65, 67, 68, . . .76 |
| 7 | Other manufacturing industries | 8, 34, 35,59, 78 |
| 8 | Constructions | 83 |
| 9 | Transports, post, and telecommunications | 87,. . .95 |
| 10 | Trade, business, and public services | 84,..., 86, 96, . . .105 |

Source: Dobrescu et al. (2010).

This operation yields two computational benefits. Firstly, branches that exhibit negative gross value added were excluded from the analysis. Secondly, this process facilitates the utilization of matrix analysis in computations.

## 4. Results

The coefficients of the A matrix calculated for 2021 can be seen in Table 2.

It is a well-established fact that inputs are sourced not only from national production but also from imports within the providing sector. Dynamic analyses have revealed that Romania's Leontief matrix coefficients exhibit a notable degree of volatility. This volatility can be attributed to the confluence of several pivotal transitional processes: the economy's structural realignment, technological advancements, and occasionally drastic fluctuations in relative prices.

**Table 2.** The input–output matrix A.

| Sector Code | 1 | 2 | 3 | 4 | 5 | 6 | 7 | 8 | 9 | 10 |
|---|---|---|---|---|---|---|---|---|---|---|
| **1** | 0.544 | 0.113 | 0.111 | 0.32 | 0.169 | 0.111 | 0.112 | 0.112 | 0.111 | 0.12 |
| **2** | 0.111 | 0.413 | 0.35 | 0.112 | 0.111 | 0.12 | 0.198 | 0.118 | 0.111 | 0.117 |
| **3** | 0.128 | 0.198 | 0.319 | 0.155 | 0.145 | 0.147 | 0.184 | 0.126 | 0.345 | 0.145 |
| **4** | 0.153 | 0.112 | 0.114 | 0.469 | 0.123 | 0.114 | 0.118 | 0.114 | 0.118 | 0.286 |
| **5** | 0.123 | 0.117 | 0.113 | 0.346 | 0.353 | 0.128 | 0.136 | 0.149 | 0.119 | 0.174 |
| **6** | 0.164 | 0.186 | 0.134 | 0.135 | 0.145 | 0.353 | 0.157 | 0.158 | 0.197 | 0.17 |
| **7** | 0.235 | 0.202 | 0.181 | 0.177 | 0.209 | 0.3 | 0.521 | 0.27 | 0.188 | 0.242 |
| **8** | 0.117 | 0.117 | 0.285 | 0.115 | 0.296 | 0.114 | 0.114 | 0.264 | 0.117 | 0.127 |
| **9** | 0.121 | 0.142 | 0.318 | 0.336 | 0.136 | 0.136 | 0.134 | 0.117 | 0.188 | 0.15 |
| **10** | 0.144 | 0.18 | 0.152 | 0.175 | 0.202 | 0.279 | 0.166 | 0.295 | 0.263 | 0.31 |

Source: own computations.

Consequently, the most recent input–output tables data (year 2021) were utilized, under the assumption that they offer a more accurate reflection of the current economic characteristics. The horizontal examination of Table 2 demonstrates that, within each sector, the most prominent coefficients are associated with self-referential inputs (values along the main diagonal). Naturally, noteworthy inputs into a sector's output manifest in scenarios such as:

- From sector 4 to sector 10;
- From sectors 2, 8, and to sector 3;
- From sectors 3 to sector 9;
- From sector 10 to sectors 6 and 8;
- From sector 8 to sector 5;
- From sectors 5 and 9 to sector 4.

The provided Leontief matrix A depicts the intersectoral relationships and coefficients of the input–output model for the ten sectors considered. Each row represents a sector's production and consumption relationships with the other sectors, as indicated by the values in the respective columns.

Upon analyzing the matrix, a few observations can be made:

Sector Interdependencies: The values in the matrix illustrate the intensity of interactions between different sectors. Sectors with higher coefficients in each row have a stronger influence on the production and output of the sector corresponding to that row.

Key Drivers: Sectors with higher coefficients on the diagonal (top left to bottom right) are pivotal drivers of their own production. These sectors have significant backward linkages, indicating that changes in their production will have considerable ripple effects on the overall economy.

Supply Chains: The matrix also highlights sectors that contribute more to the inputs of other sectors (higher values in the corresponding columns). This implies that these sectors play a crucial role in the supply of intermediate goods and services to other sectors for their production processes.

Potential Multiplier Effects: The off-diagonal coefficients provide insight into the potential multiplier effects of investments or changes in final demand. Alterations in the final demand for a particular sector can lead to cascading effects across other sectors through supply chain relationships.

Structural Characteristics: The distribution of coefficients across the matrix reflects the structural characteristics of the economy. Sectors with larger coefficients in certain rows or columns may indicate the relative importance of these sectors in the overall economic landscape.

Complexity of Interactions: The varying coefficient values illustrate the complexity of intersectoral relationships. Sectors with high coefficients in multiple rows or columns may have intricate linkages and dependencies with various parts of the economy.

Overall, this input–output analysis serves as a valuable tool for understanding how changes or shocks in one sector can reverberate through the entire economy, influencing production, output, and economic growth. By interpreting the coefficients and relationships of the matrix, policymakers, economists, and analysts can make informed decisions to promote balanced economic development and stability.

Since the cross-sector exchange table is built in prices, the coefficients $a_{ij}$ can sum up vertically (columns) ($S_c$) and horizontally (rows) ($S_r$), they can be seen in the following Table 3:

**Table 3.** Total sum of coefficients by vertical and horizontal addition.

| Sector Code | Sr | Sc |
| --- | --- | --- |
| 1 | 1.823 | 1.84 |
| 2 | 1.761 | 1.78 |
| 3 | 1.692 | 1.907 |
| 4 | 1.641 | 1.94 |
| 5 | 1.558 | 1.709 |
| 6 | 1.799 | 1.702 |
| 7 | 2.725 | 1.84 |
| 8 | 1.316 | 1.663 |
| 9 | 1.378 | 1.557 |
| 10 | 2.006 | 1.761 |

Source: own computations.

The sum of column values (Sc) serves as an estimation of the proportion of intermediate consumption within the sector's output. Deviations from unity approximate the proportion of gross value added to the output. It is worth noting that, uniformly, the values of Sc are less than one, thus corroborating that, within the applied aggregation, all sectors exhibit positive gross value added. There are discernible variations in the magnitudes of Sc among sectors.

The summation of rows (Sr) results provides an approximation, indicated by deviations from unity, of the relative contribution of domestic output in meeting the economy's intermediate consumption. Sector 7 (other manufacturing industries) exhibits the most pronounced deficit in this aspect, pointing to a considerable reliance on imports for industries employing materials and semi-finished goods. Conversely, sector 8 (constructions) stands out as a counterpoint, primarily serving significant segments of final demand such as residential and productive investment, and infrastructure projects.

Table 3 provides a comprehensive depiction of the total coefficients' aggregation achieved via both vertical (column) and horizontal (row) summation for each sector in the analyzed Leontief matrix. This analysis serves to illuminate the interconnected relationships among sectors and the overall significance of diverse sectors within the economy.

By examining the results, several key observations can be made:

Vertical and Horizontal Sums: The values in the "Sr" and "Sc" columns represent the total sum of coefficients obtained by adding either vertically (columns) or horizontally (rows) across each sector. These sums provide insight into the cumulative impact of each sector on other sectors and on the economy.

Sector Significance: Sectors with higher total coefficients are more impactful within the economy, as indicated by their larger sums. These sectors have substantial linkages and dependencies with other sectors and play a critical role in the overall economic network.

Sector Balance: Comparing the "Sr" and "Sc" values for each sector allows us to assess the balance of relationships. If the "Sr" and "Sc" values are relatively close, this suggests a well-balanced distribution of inputs and outputs within the sector and its interactions with others.

Differential Impact: Discrepancies between the "Sr" and "Sc" values highlight sectors with differing levels of influence as suppliers and consumers. Sectors with higher "Sr" values indicate that they contribute more to other sectors' inputs, while sectors with higher "Sc" values rely more heavily on inputs from other sectors.

Intersectoral Connectivity: Sectors with significant differences between their "Sr" and "Sc" values may play pivotal roles in connecting various parts of the economy. Such sectors act as intermediaries, facilitating the flow of inputs and outputs across different sectors.

Economic Dynamics: Variation in the total coefficients' sums across sectors reflects the dynamic nature of the economy. Sectors with higher "Sr" and "Sc" values may be subject to stronger feedback loops, amplifying their impact on the overall economic performance.

In summary, Table 3 offers a valuable glimpse into the interconnectedness and relative importance of sectors within the economy. The total sum of coefficients, obtained through both vertical and horizontal addition, provides a quantitative measure of each sector's contribution to the overall economic activity and highlights sectors that play crucial roles in facilitating intersectoral relationships and economic growth.

The inverse matrix $(I - A)^{-1}$ holds analytical significance as a numerical representation of interconnections within the economy's sectors. These connections encompass both direct relationships, as indicated in matrix A, and additional indirect interdependencies.

When expressed in terms of prices, the vertically aligned coefficients, as observed in the context of the matrix $(I - A)^{-1}$, can also be aggregated, denoted as Sc. These aggregates provide an approximation of the collective output generated across all industrial branches in response to a unit of final demand directed toward the specific sector corresponding to the given vertical (thus, in the matrix $(I - A)^{-1}$, coefficients on the primary diagonal exhibit values exceeding unity). The summation of the vertical coefficients within the matrix $(I - A)^{-1}$ is commonly referred to as output multipliers, a term introduced by (Miller and Blair 2009). This terminology underscores the broader economic implications of sector-specific demands on overall output.

Upon analyzing the matrix, several noteworthy observations can be made:

Cross-Sector Interactions: The coefficients in the matrix provide a quantifiable measure of the interactions between different sectors. Larger values suggest stronger interdependencies, indicating that changes in one sector can have significant ripple effects throughout the economy.

Indirect Effects: The coefficients demonstrate not only direct effects (represented by the diagonal) but also indirect effects that propagate through the economy. Sectors with higher indirect coefficients highlight their role in transmitting influence on other sectors.

Sector Contributions: Sectors with higher coefficients indicate their elevated contribution to the economy's overall intersectoral dynamics. These sectors play a pivotal role in shaping the economic network.

Influence Amplification: The matrix coefficients demonstrate how changes in one sector can magnify their impact as they reverberate through other sectors. This amplification underscores the complex web of relationships within the economy.

Network Structure: The matrix offers a visual representation of the structure of sectoral interactions. Sectors with pronounced interdependencies can act as central nodes, connecting various parts of the economy.

Comparative Analysis: Comparing the values between rows and columns provides insight into how sectors contribute and rely on each other. Variations in coefficients highlight the differing roles of sectors in the economy's overall functioning.

In conclusion, Table 4 provides a comprehensive portrayal of the matrix coefficients of $(I - A)^{-1}$, revealing the intricate web of intersectoral relationships within the economy.

This analysis facilitates a deeper understanding of how changes and influences in one sector cascade through the economy, shaping its overall behavior and performance.

**Table 4.** The matrix coefficients of $(I - A)^{-1}$.

| Sector Code | 1 | 2 | 3 | 4 | 5 | 6 | 7 | 8 | 9 | 10 |
|---|---|---|---|---|---|---|---|---|---|---|
| 1 | 1.371 | 0.379 | 0.451 | 0.099 | 0.204 | 0.344 | 0.409 | 0.316 | 0.278 | 0.290 |
| 2 | 0.405 | 1.111 | 0.047 | 0.461 | 0.327 | 0.283 | 0.178 | 0.290 | 0.271 | 0.337 |
| 3 | 0.323 | 0.163 | 0.951 | 0.326 | 0.239 | 0.221 | 0.199 | 0.244 | 0.196 | 0.253 |
| 4 | 0.224 | 0.292 | 0.345 | 1.214 | 0.232 | 0.260 | 0.308 | 0.236 | 0.195 | 0.115 |
| 5 | 0.263 | 0.247 | 0.294 | 0.256 | 1.087 | 0.203 | 0.235 | 0.155 | 0.175 | 0.151 |
| 6 | 0.274 | 0.204 | 0.320 | 0.366 | 0.252 | 1.014 | 0.274 | 0.218 | 0.139 | 0.238 |
| 7 | 0.505 | 0.482 | 0.281 | 0.694 | 0.454 | 0.285 | 0.927 | 0.327 | 0.397 | 0.441 |
| 8 | 0.192 | 0.171 | 0.205 | 0.222 | 0.157 | 0.160 | 0.192 | 1.026 | 0.128 | 0.157 |
| 9 | 0.220 | 0.163 | 0.222 | 0.227 | 0.158 | 0.155 | 0.189 | 0.169 | 0.928 | 0.154 |
| 10 | 0.366 | 0.261 | 0.351 | 0.362 | 0.209 | 0.239 | 0.308 | 0.151 | 0.094 | 0.892 |

Source: own computations.

The $(S_r)$ values resulting from matrix A do not involve the second quadrant of the input–output table. To make these values somewhat comparable to the corresponding vector $(S_c)$, we use the relation $(S_r)^* = (S_r) + 1$. The two sets of values $(S_c)$, and $(S_r)^*$ are shown in Table 5.

**Table 5.** Total sum of coefficients by vertical and horizontal addition.

| Sector Code | Sc | Sr* |
|---|---|---|
| 1 | 2.771 | 2.823 |
| 2 | 3.304 | 2.761 |
| 3 | 2.792 | 2.692 |
| 4 | 3.198 | 2.641 |
| 5 | 2.802 | 2.558 |
| 6 | 3.023 | 2.799 |
| 7 | 4.288 | 3.725 |
| 8 | 2.416 | 2.316 |
| 9 | 2.365 | 2.378 |
| 10 | 2.866 | 3.006 |

Source: own computations. $(S_r)^* = (S_r) + 1$.

The difference between $(S_c)$ and $(S_r)^*$ is attributable to the way the effect of productive interdependencies within the economy is expressed. While $(S_r)^*$ is limited to the direct ones, Sc adds to them the indirect interdependencies (mediated by the links between the related branches).

Table 5 offers a comprehensive view of the total coefficients derived from vertical and horizontal summation, revealing the interconnected relationships between sectors and their impact on the overall economy.

The "Sc" column signifies the collective influence of sectors on other sectors, depicting their role in providing intermediate goods and services. Sectors with higher "Sc" values act as significant suppliers within the economic network.

Conversely, the "Sr*" values in the horizontal summation indicate the relative importance of sectors in driving overall economic activity. Sectors with larger "Sr*" values play pivotal roles in generating output and contributing to the economy's growth.

Comparing the "Sc" and "Sr*" values offers insights into the dual contributions of sectors—as suppliers and as drivers of economic output. The variations in "Sc" and "Sr*" values underscore the complex web of interdependencies among sectors, influencing their susceptibility to changes in demand and supply.

We can now compute an index representing the indirect influence on output, which is calculated as the ratio $\gamma = (S_c)/(S_r)^*$, as illustrated in Figure 1. In the context of the $(I - A)^{-1}$ matrix, the summation of coefficients along the horizontal (sr) axis provides an estimate of the output needed by each sector to accommodate a unit increase in final demand across all sectors of the economy.

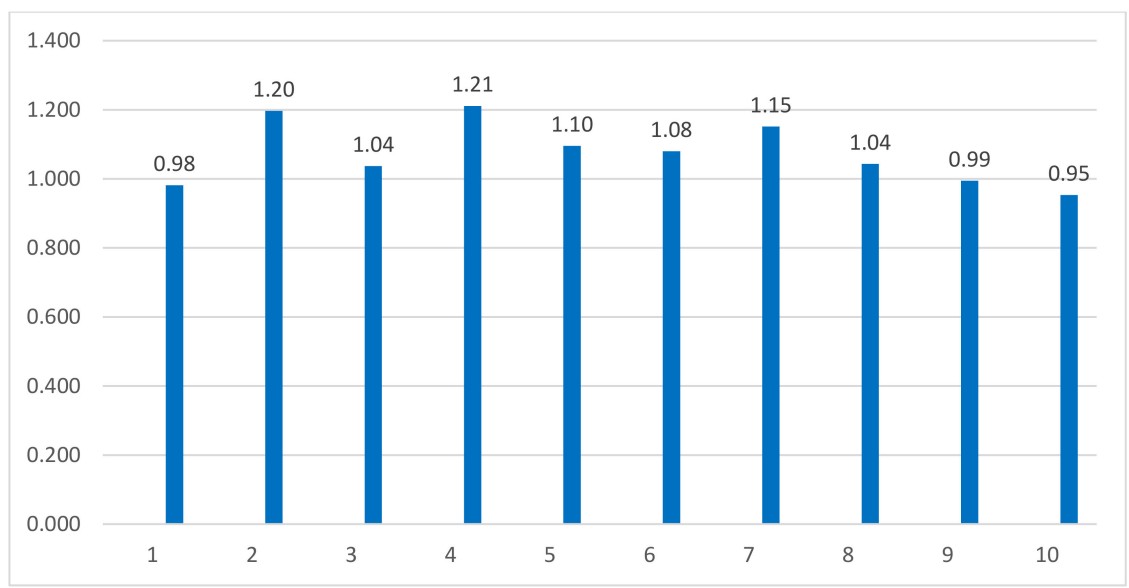

**Figure 1.** The values of w, by sector. Source: own computations.

It is important to highlight that comparing these values across matrices of different sizes would not yield meaningful comparisons. To address this aggregation effect, dividing these values by the number of sectors can normalize them, resulting in values associated with a unit change in the final demand within the economy. Nonetheless, it is important to exercise caution when utilizing these values for explanatory purposes, as they are based on the simplifying assumption of uniform modifications in final demand across all sectors, which may not accurately reflect real-world dynamics.

The above-presented Figure 1 displays a set of results representing different sectors, each denoted by a specific sector code, along with corresponding values labeled as "w". These values indicate a scaling factor or coefficient associated with each sector, suggesting a certain level of influence or impact.

Several observations can be drawn from the results:

Variability in Impact: The "w" values exhibit variability across different sectors, indicating distinct levels of influence or contribution. Sectors with higher "w" values are likely to have a more pronounced impact on certain economic measures.

Magnitude of Effect: The numerical values of "w" provide insights into the extent of the impact that each sector exerts. Higher values suggest a stronger influence, while lower values indicate a relatively lower impact.

Relative Importance: The values in the table allow for the comparison of sectors in terms of their contributions or significance. Sectors with larger "w" values are relatively more important within the context under consideration.

Potential Policy Implications: These results can guide policymakers and analysts in making informed decisions by identifying sectors with higher or lower impacts. Policy initiatives could be tailored to leverage sectors with higher "w" values for greater economic benefits.

Complex Determinants: The factors determining the "w" values can involve a combination of factors such as sector size, intersectoral relationships, supply chain dynamics, and demand patterns.

Consideration of Context: While the "w" values provide a quantitative measure of sectoral influence, their interpretation should consider the broader economic context, including the objectives of the analysis and the specific variables being considered.

In summary, the provided table of "w" values offers insights into the relative influence or impact of different sectors within a certain context, enabling a better understanding of sectoral dynamics and potential implications for policy and decision-making.

The research findings presented in the tables offer critical insights with direct policy relevance, particularly in the context of economic development, investment strategies, and supply chain management. Highlighting these policy-relevant results can effectively hook the reader and inform policymakers of their implications:

Sectoral Investment Strategies: The input–output analysis in Table 2 underscores the significance of sector-specific investment strategies. Policymakers can use this data to tailor investment incentives and support for sectors with strong interdependencies, thereby promoting economic growth and job creation. For example, identifying sectors with high backward linkages can guide targeted investment in research and development to stimulate innovation.

Economic Resilience: The complex determinants of the "w" values, as seen in Figure 1, can serve as a policy compass for enhancing economic resilience. Sectors with higher "w" values can be considered as economic anchors that play a critical role in overall economic stability. Policymakers can prioritize these sectors for strategic investments and disaster preparedness, ensuring a robust economic backbone.

Supply Chain Diversification: The results highlight the importance of diversifying supply chains to minimize reliance on specific sectors, reducing vulnerability to external shocks. Policymakers can use this insight to promote diversification efforts and strengthen domestic production capacities, thereby enhancing economic resilience and reducing the impact of global disruptions.

Balanced Regional Development: Regional development policies can leverage the input–output analysis findings to promote balanced development across regions. Focusing on sectors with high interdependencies can help policymakers design region-specific strategies to foster economic growth and reduce regional disparities.

Investment Allocation: The observed variations in the matrix coefficients indicate that not all sectors have equal importance in the economy. Policymakers can optimize resource allocation by considering the impact of different sectors on the overall economy. This information can guide investment priorities and facilitate better resource management.

Global Trade Strategies: Understanding the sector-specific interdependencies, as revealed by the input–output analysis, can inform trade strategies. Policymakers can negotiate trade agreements and partnerships with an awareness of their potential impact on various sectors. This approach can mitigate potential risks and maximize economic benefits.

Fostering Innovation: Sectors with high coefficients in the input–output analysis are integral drivers of economic growth. Policymakers can harness this information to foster innovation in these sectors through research grants, incentives, and education programs, ultimately leading to increased productivity and competitiveness.

### 4.1. Exploring Prior Evidence

Before delving into the specifics of our findings, it is prudent to contextualize the results within the existing body of knowledge. The field of economics has seen a proliferation of research dedicated to understanding the relationship between investments

and economic performance. Notably, this relationship is often contingent on the unique economic characteristics of a given nation, making it imperative to explore the nuances of Romania's economic landscape.

Numerous empirical studies have highlighted the pivotal role of investments as drivers of economic growth and development. A consistent finding across these studies is that investments, whether in physical capital, human capital, or research and development, yield multifaceted benefits. They not only enhance productivity but also stimulate innovation, boost efficiency, and create a conducive environment for business expansion. In alignment with these existing insights, our study reveals similar dynamics within the Romanian economy.

One aspect that our findings resonate with is the importance of the level of economic development in shaping the outcomes of investments. Developed and developing economies exhibit distinct patterns in their investment dynamics. In the context of Romania, a transitioning, emerging economy, our results reflect a landscape where investments play a critical role in bridging infrastructure gaps, fostering technological capabilities, and stimulating growth. This is in line with prior evidence suggesting that investments can serve as catalysts for emerging markets by addressing structural constraints and enabling efficient resource allocation.

Moreover, the literature emphasizes that the impact of investments is not a unilateral process. The external economic environment, characterized by factors such as global trade dynamics, economic shocks, and market uncertainties, exerts a profound influence on how investments translate into economic output. Considering this, our study corroborates the importance of considering external factors in the investment–performance nexus. Our findings echo previous research by showcasing the intricate ripple effects that external disruptions can introduce, necessitating adaptive strategies and policy frameworks.

*4.2. Contributions and Implications*

Building on the foundations of prior evidence, our study introduces valuable insights into the specific case of the Romanian economy. The nuanced relationships between investments, economic development stages, and external influences are revealed within this context. This deeper understanding opens the door to policy adjustments, investments, and strategies that are tailored to Romania's unique economic dynamics.

Our results underscore the significance of investments in shaping the multifaceted facets of economic performance. Whether it be augmenting physical capital, fostering innovation, or addressing infrastructure gaps, investments play a pivotal role in the economic evolution of Romania. This realization carries direct policy implications. Policymakers in Romania can harness these insights to structure investment initiatives that target specific areas of the economy, aiming to catalyze growth and development effectively.

In conclusion, our study's findings not only corroborate the established link between investments and economic performance but also add the critical layer of specificity by analyzing these dynamics within the Romanian context. By bridging the gap between existing knowledge and the unique characteristics of Romania's economic landscape, we offer a stepping stone for evidence-based decision-making. Romania, as a transitioning economy, stands to benefit from tailored investments and strategies that can foster sustainable growth and innovation, ultimately positioning the nation on a path toward economic resilience and prosperity.

## 5. Conclusions

Through a meticulous examination of input–output tables for the Romanian economy, we have discerned the intricate web of interdependencies that connect various sectors. Our findings underscore the significance of investments as catalysts for economic growth and development, with distinct implications contingent upon the type of investment, the level of development, and external factors.

We observed that different types of investments, ranging from physical capital to research and development, play pivotal roles in shaping productivity, innovation, and efficiency. The level of development of a country introduces additional complexities, and developing economies reap substantial benefits from investments that bridge infrastructure gaps and enhance technological capabilities. Conversely, advanced economies navigate a nuanced landscape where efficiency gains, market saturation, and quality of investment allocation assume greater importance.

External factors, such as trade conflicts and global shocks, have a profound impact on investment dynamics and subsequent economic output. The intricate ripple effects of these external disruptions necessitate adaptive strategies and policy frameworks to navigate the resulting uncertainties, findings showing the same evidence as (Özker 2021).

Our exploration of the input–output analysis has provided insight into the quantifiable interdependencies among sectors, unraveling the intricate supply chain networks and multiplier effects that underpin economic growth. The computed indices of output indirect drive further contribute to our understanding of sectoral interplay, offering a quantitative lens through which the economy's responsiveness to changes in demand can be examined.

### 5.1. Summary of Findings

In summarizing our research findings, we highlight the intricate relationship between investments and economic performance within the Romanian context. Our study reveals that investments, whether in physical capital, research and development, or infrastructure, play pivotal roles in influencing productivity, innovation, and efficiency. The level of development, a critical factor in shaping the outcome, introduces complexities that distinguish between emerging and developed economies.

External factors, such as global shocks and trade dynamics, have a profound impact on the investment–performance nexus. Our analysis elucidates the ripple effects of these external influences, underscoring the need for adaptive strategies to navigate uncertainties. The input–output model, a powerful analytical framework, provides a quantitative lens through which the economy's responsiveness to changes in demand can be examined.

Our findings are rooted in data collected from the most recent input–output tables of the Romanian economy, representing ten sectors. This comprehensive analysis quantifies the interdependencies among sectors, offering insights into sectoral contributions, supply chain dynamics, and potential multiplier effects. In a dynamic and complex economic landscape, our study contributes to the ongoing discourse on the intricate relationships that underlie economic development.

### 5.2. Limitations

It is essential to acknowledge the limitations inherent in our analysis. We assumed a simplified scenario of uniform modification in final demand across sectors, which may not always align with real-world scenarios. The aggregation effect in comparing matrices of varying sizes underscores the need for cautious interpretation.

The scope of our study is constrained to the Romanian economy and specific input–output tables from 2021. Generalizability to other economies and time periods may be subject to limitations based on variations in economic structure, development stages, and external factors.

### 5.3. Future Directions

Future research endeavors can build upon this study to address these limitations and explore uncharted territories. Investigating non-linear relationships and incorporating time-dependent factors could enhance the accuracy of predictive models. Expanding the study to encompass international trade dynamics and global supply chains could yield a more holistic understanding of how investments impact global output.

Exploring the role of technological advancements and digital transformations in investment-induced output changes could offer insights into the evolving nature of eco-

nomic growth. Advanced econometric techniques and machine learning methodologies could be harnessed to analyze intricate interdependencies with greater precision.

Moreover, expanding the research to encompass a broader range of economic indicators, such as employment, inequality, and environmental sustainability, could provide a more comprehensive evaluation of the multifaceted effects of investments on the economy.

*5.4. Final Remarks*

In conclusion, while our study has enriched our understanding of the impact of investments on global output, it lays the groundwork for future investigations to address the limitations outlined and delve deeper into the complexities of economic interactions. By embracing a multidimensional approach and accounting for evolving economic landscapes, future research endeavors can contribute to a more nuanced comprehension of investment dynamics and their profound implications for sustainable economic growth and development. As the global landscape evolves, this study serves as a valuable resource for shaping strategies that foster sustainable growth, innovation, and resilience in an ever-changing economic environment.

**6. Recommendations**

Based on the analysis of the Leontief input–output model for Romania, the following policy recommendations can be considered:

1.　　Prioritize investments in key industries

Focus on targeted investments in sectors that have a high potential for growth and can generate significant spillover effects across the economy. These sectors could include:

Information technology and communication (ICT): Promote digital transformation and enhance Romania's ICT infrastructure to support innovation, improve productivity, and attract foreign investment.

Renewable energy: Encourage the development of renewable energy sources such as solar, wind, and geothermal power to reduce reliance on fossil fuels, enhance energy security, and mitigate climate change impacts.

Tourism: Invest in tourism infrastructure, promote Romania's cultural heritage and natural beauty, and develop innovative tourism packages to attract more visitors and boost the tourism industry.

2.　　Implement targeted subsidies and incentives

Provide strategic subsidies and incentives to encourage innovation, increase productivity, and promote the adoption of new technologies. These measures could include:

Research and development (R&D) subsidies: Support investments in research and development to drive innovation and promote technological advancements across various sectors.

Skill development and training programs: Invest in upskilling and reskilling programs to equip the workforce with the necessary skills to meet the demands of the evolving economy.

Tax incentives for green investments: Provide tax breaks and other incentives to encourage businesses to invest in renewable energy technologies, energy-efficient practices, and sustainable production processes.

3.　　Optimize government spending

Ensure that government spending is aligned with economic priorities and maximizes its impact on growth and development. This could involve:

Investing in infrastructure: Allocate resources for infrastructure development, including transportation networks, energy infrastructure, and telecommunications infrastructure, to improve connectivity, reduce transportation costs, and enhance productivity.

Supporting education and healthcare: Invest in education and healthcare to improve human capital, enhance workforce skills, and promote overall well-being.

Promoting entrepreneurship and innovation: Provide funding, mentorship, and support programs for startups and small businesses to foster innovation, create jobs, and drive economic growth.

4. Implement strategic tariffs and trade policies

Utilize tariffs and trade policies judiciously to protect domestic industries, promote fair competition, and encourage the development of strategic industries. This could involve:

Protecting strategic industries: Implement temporary tariffs on imported goods to safeguard domestic industries while they mature and become competitive in the global market.

Negotiating trade agreements: Pursue trade agreements with key partners to expand market access for Romanian products, attract foreign investment, and promote economic integration.

Promoting export competitiveness: Provide support and incentives to Romanian businesses to enhance their competitiveness in export markets.

By implementing these policy recommendations, Romania can effectively leverage its resources, strengthen its economic competitiveness, and promote sustainable economic growth. The Leontief input–output model provides valuable insights into the interdependencies between sectors and the impact of policy decisions, allowing policymakers to make informed choices that benefit the Romanian economy.

5. Domestic Distortions: A Critical Aspect of Economic Growth

While investments play a crucial role in driving economic growth, their effectiveness can be hindered by various domestic distortions that permeate the Romanian economy. These distortions, such as wage inflexibility, credit market imperfections, and structural rigidities, can impede the efficient allocation of resources, stifle innovation, and limit the overall potential for economic growth.

Wage inflexibility, a prevalent issue in Romania, arises from rigid labor market regulations that prevent wages from adjusting to market conditions. This can lead to labor market mismatches, as firms may struggle to attract and retain skilled workers at competitive wages. Moreover, wage inflexibility can hinder firms' ability to adjust to economic shocks, potentially exacerbating unemployment and economic stagnation.

Credit market imperfections, another significant distortion in Romania, hinder the flow of credit to businesses, particularly small- and medium-sized enterprises (SMEs). This can limit investment opportunities, restrain SMEs' growth potential, and contribute to a constrained overall economy. Additionally, structural rigidities, such as inefficient government procurement processes and bureaucratic hurdles, can further impede investment and economic growth. These rigidities can discourage foreign investment and make it challenging for domestic businesses to operate efficiently.

To effectively address these domestic distortions and promote sustainable economic growth, policymakers need to adopt a comprehensive and well-targeted approach. This includes measures to enhance labor market flexibility, improve credit market access, and streamline regulations to foster a more business-friendly environment.

6. Wage Inflexibility and Its Impact on Unemployment

One of the significant domestic distortions that can hinder economic growth in Romania is wage inflexibility. Rigid labor market regulations that prevent wages from adjusting to market conditions can lead to several adverse consequences.

Firstly, wage inflexibility can create labor market mismatches, where the demand for and supply of labor are out of sync. This can result in skill shortages and an oversupply of labor in certain sectors, leading to unemployment and underemployment.

Secondly, wage inflexibility can limit firms' ability to adjust to economic shocks, such as changes in demand or productivity. During economic downturns, firms may be reluctant to reduce wages, even if it is necessary to remain competitive and avoid layoffs. This can prolong periods of economic stagnation and exacerbate unemployment.

Thirdly, wage inflexibility can distort resource allocation and hinder innovation. If wages are not responsive to market signals, firms may be less inclined to invest in new technologies or training programs, as they may not expect to reap the full returns from these investments. This can stifle productivity growth and limit the overall potential for economic expansion.

To address wage inflexibility and promote sustainable employment, policymakers need to adopt a balanced approach that encourages labor market flexibility while ensuring fair compensation for workers. This includes measures such as reforming labor market regulations, promoting collective bargaining, and supporting skills development programs.

7.    Lessons from Bhagwati, Johnson, and Corden: A Framework for Optimal Policies

The works of eminent economists Jagdish Bhagwati, Harry Johnson, and Max Corden provide valuable insights into the selection of optimal policies for addressing domestic distortions and promoting economic growth. Their emphasis on the need for comprehensive and well-targeted policies resonates strongly with the challenges faced by the Romanian economy.

Bhagwati, Johnson, and Corden advocate for carefully evaluating the costs and benefits of various policy options before implementation. They emphasize that tariffs and other trade policies should be used judiciously, as they can have unintended consequences and potentially hinder economic efficiency.

Jagdish Bhagwati is renowned for his advocacy of free trade and his critique of protectionist policies. He highlights how domestic distortions such as tariffs and trade barriers hinder economic growth and efficiency. Bhagwati (1969) emphasizes that while protectionist measures might aim to shield domestic industries, they often lead to market inefficiencies, reduced competitiveness, and an overall decline in consumer welfare. His work underscores the importance of adopting policies that promote openness in international trade, leading to increased efficiency and overall welfare gains.

Harry Johnson's contributions to the analysis of domestic distortions (Johnson 1963) lie in his extensive research on exchange rates, balance of payments, and their implications for economic policies. He emphasizes the interconnectedness of domestic and international economic factors. Johnson's insights into exchange rate policies shed light on how these policies can inadvertently create distortions within domestic markets. His research aids in understanding the complexities of policy formulation to counter these distortions and achieve optimal economic outcomes.

Max Corden's work (1957) focuses on the effects of trade policies, particularly tariffs and subsidies, on economic welfare. Corden's analysis illustrates how domestic distortions arising from trade policies can significantly impact trade patterns and economic development. He emphasizes the need for careful consideration of policy measures, suggesting that while protective policies might offer short-term benefits to specific industries, they often lead to long-term inefficiencies and welfare losses.

Collectively, these economists provide a robust framework for policymakers to evaluate and select optimal policies to address domestic distortions. Their insights underscore the necessity of conducting comprehensive cost–benefit analyses when considering alternative policies. By weighing the short-term gains against the long-term costs, policymakers can make informed decisions that minimize distortions, maximize economic efficiency, and enhance overall welfare. Bhagwati, Johnson, and Corden's extraordinary work serves as a guiding beacon for policymakers seeking to navigate the complexities of domestic distortions and formulate policies that foster sustainable economic growth and prosperity.

In the context of Romania, this framework suggests that policymakers should carefully assess the potential impact of any proposed policy on domestic distortions and overall economic growth. Targeted interventions, such as reforms to labor market regulations, financial market reforms, and regulatory streamlining, could be more effective in addressing the root causes of economic stagnation than relying solely on tariffs or other trade barriers.

The decision to model the Romanian economy as a closed economy in this analysis is based on several considerations. First, the primary purpose of this analysis is to focus



on the direct and indirect effects of investments on domestic output and employment. By assuming a closed economy, we can isolate these effects from the complexities associated with international trade and finance.

Second, the Romanian economy has a relatively high degree of domestic production, with a significant portion of goods and services consumed within the country. This domestic orientation suggests that the impact of investments on economic growth is more pronounced within the domestic economy, making a closed economy assumption more relevant.

Third, the analysis draws on input–output data that provides a detailed representation of the interdependencies between industries within the Romanian economy. By assuming a closed economy, we ensure that the input–output data are consistent with the model and accurately reflect the structure of the domestic economy.

Fourth, the policy recommendations proposed in the analysis are primarily targeted at domestic policymakers, focusing on measures that can be implemented within the Romanian economy to promote economic growth. A closed economy assumption allows for more tailored policy recommendations that are directly applicable to the domestic context.

Finally, while a closed economy assumption simplifies the analysis and focuses on domestic factors, it is important to acknowledge its limitations. International trade and finance play a significant role in the Romanian economy, and their exclusion from the model can lead to some degree of distortion. Future research could incorporate more complex models that consider the impact of international trade and finance on the Romanian economy.

**Author Contributions:** Conceptualization, M.B., M.V.V. and S.A.; methodology, M.B.; software, M.B.; validation, M.B., M.V.V. and S.A.; formal analysis, M.B.; investigation, M.V.V.; resources, S.A.; data curation, M.V.V.; writing—original draft preparation, M.B.; writing—review and editing, M.V.V.; visualization, S.A.; supervision, M.V.V. and S.A.; project administration, S.A. All authors have read and agreed to the published version of the manuscript.

**Funding:** This research received no external funding.

**Data Availability Statement:** Publicly available datasets were analyzed in this study. This data can be found here: [https://insse.ro/cms/en/tags/national-accounts].

**Acknowledgments:** This work was supported by a grant from the Romanian Ministry of Education and Research, CNCS-UEFISCDI, project number PN-III-P4-ID-PCE-2020-0557, within PNCDI III, contract number 112/2021.

**Conflicts of Interest:** The authors declare no conflict of interest.

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
