# Peer review of "Navigating the Intricate Relationship between Investments and Global Output: A Leontief Matrix Case Study of Romania"

_jrfm, doi:10.3390/jrfm16120521_

Round 1

Reviewer 1 Report (Previous Reviewer 1)

Comments and Suggestions for Authors

Dear authors,
After the first reviews, I observe the great effort in highlighting the contribution. In its current state, it is interesting to the reader and the methodology used is clarified.
Congratulations for the good work!

Author Response

Dear professor,

Thank you for all your help during the revision process.

Best regards

Reviewer 2 Report (Previous Reviewer 3)

Comments and Suggestions for Authors

The English and grammar in this paper is fine. There are two things that need to be done in order to make it publishable. The model needs to be explained intuitively. i want to be able to understand the paper without working through the mathematical model. The authors need to explain in words what the assumptions are in the model and what the experiment is behind the multipliers calculated. 

Second i need to know precisely what the policy recommendations are for Romania. Subsidies? Taxes? Government spending? tariffs? I hope the authors will follow up on these suggestions. I look forward to reading the next draft.

Author Response

Dear professor,

Thank you for all your valuable comments. Please see attached our responses to your review.

Best regards

Reviewer 3 Report (New Reviewer)

Comments and Suggestions for Authors

Very interesting research, I support its publication.

Author Response

Dear professor,

Thank you for all your valuable comments.

Best regards.

Reviewer 4 Report (New Reviewer)

Comments and Suggestions for Authors

In principle:

In the part written in black, the paper content, organisation and methodology completely follows Dobrescu, Emilian and Gaftea, Viorel Nicolae and Scutaru, Cornelia, Using the Leontief matrix to estimate the impact of investments on global production (12 April 2010). Romanian Economic Forecast Journal, Vol. 13, No. 2, pp. 176-187, 2010, available on SSRN: https://ssrn.com/abstract=1635743.

This reference to Dobrescu et al is not quoted. Countrary, another reference to Dubrescu (2009) seems to no be as pertinent.

The unique contribution comes from the Data, that seem to be upgrated.

Also the paper’s title is very similar.

Leontief is never quoted in the references list

In the additional part in red, some sentences are not well understandable. They seem a no required justification about gthe authors' work.

The reference to “resilience” and “supply chain” is not clear in the framework of the Leontief’s approach, also if it could be an interesting aspect of the issue’s evolution.

The paper in the whole is devoted to the Romania's case. This not coherent ith the title and the abstract, where the country is presented as a focus of a more general analysis  on informing "policy decisions  and drive sustainable growth in an increasingly complex global economy".

Romania word is not in the keywords

Author Response

Dear professor,

Thank you for all your valuable comments. They helped us to improve our manuscript.

Please see attached our responses.

Best regards.

Round 2

Reviewer 2 Report (Previous Reviewer 3)

Comments and Suggestions for Authors

I wrote in my earlier report that i did not understand the paper. This is not because i don't understand input output analysis. In fact i am a coauthor of Leontief. You recommend policy recomendations including tariffs. But tariffs are seldom first best policies to deal with an economy's distortions. You have not mentioned any distortions or problems like wage inflexibility leading to unemployment. You need to imbed the analysis in a coherent macro framework. What are your distortions? Why is it rational to assume a closed economy? Is there some reason why you are modeling the Rumanian economy as a closed economy? As it stands I think this paper needs more work to be useful to policy makers in Rumania. I would like to see you address these questions, and at least have some discussion about domestic distortions discussed by Bhagwati, Johnson, and Corden. I mention them as examples of people who have done extraordinary work on the selection of optimal policies and cost benefit analysis of alternative ways to deal with distortions. I hope you will recouch the paper in terms of optimal policies and structure it so that policy makers are more knowledgeable after reading it.

Author Response

Dear professor,

Thank you for all your valuable comments. They helped us improve our manuscript significantly. 

Please see attached our responses.

Best regards.

Reviewer 4 Report (New Reviewer)

Comments and Suggestions for Authors

The review made better the paper, which could be a good research issue in order to implement the macroeconomic models towards the global interpretation of economy. I suggest to A. to be more propositive and innovative in managing this topic.

Author Response

Dear professor,

Thank you for all your valuable suggestions. They helped us improve our paper.

Best regards.

Round 3

Reviewer 2 Report (Previous Reviewer 3)

Comments and Suggestions for Authors

The paper has been improved markedly. I believe that the way to handle the problem that the authors address is a full fledged general equilibrium cost benefit analysis of alternative interventions with distortions fully accounted for in the model. I hope that in future work the authors will do this. I would like the authors to think about one more section: namely how to build a consistent model to provide cost benefit analysis of what they consider an important policy intervention. If they added it to the paper that would be good but I do not insist on it.

This manuscript is a resubmission of an earlier submission. The following is a list of the peer review reports and author responses from that submission.

Round 1

Reviewer 1 Report

Comments and Suggestions for Authors

Dear authors,
Below I present my comments and suggestions about the work. Although the a priori idea could be original, it needs a lot of work and robustness to become a publishable work. I will try to summarize the points for improvement that I have detected.
1- In the introduction-motivation the authors do not use any solid reference that supports the interest. It gives the impression that by seeking to apply a methodology a context is described. But this is not a good motivation. It is recommended to do a good review, detect the state of the frontier of knowledge on this topic, and once solved, try to contribute.
2- In my opinion, the review of the literature has two serious problems. The first is that high-impact finance journals are not used (see first quartile in JCR) and the references are somewhat obsolete, 2009 etc... A review of this methodology in a similar context would be appropriate.
3- The methodology, based on the Leontief Matrix, is not justified. That is, why this methodology can present different results and why its use is more appropriate than others.
4- The results are simply discussed but at no time is there a deep discussion regarding other previous evidence.

Major comment: It seems that a methodology that is known from another context is applied, but the financial foundations are very scarce. I hope that the comments serve to focus the work in a different way.
Good luck

Comments on the Quality of English Language

Must be improved

Reviewer 2 Report

Comments and Suggestions for Authors

The introduction provides a comprehensive overview of the economic impact of investments on global output, especially in the Romanian context. To improve, it could benefit from a clearer structural flow, starting with a general statement on investment's importance. Delving deeper into specific examples for various investment types, giving more context on Romania's relevance, expanding on external factors' effects, and incorporating additional foundational references would enhance its depth and clarity.

Enhancing the dataset by integrating qualitative interviews with key stakeholders could provide deeper insights into the nuanced impacts of investments in the Romanian economy.

To enhance the study's methodology, it would be beneficial to integrate data across multiple years for a temporal analysis, capturing the evolving influence of investments. Additionally, employing data-driven techniques like clustering algorithms for sector aggregation might offer fresh insights into sectoral interactions and dependencies, beyond the traditional classifications referenced in prior literature.

The conclusion offers a detailed understanding of the study's findings, methodologies, limitations, and future research prospects. To enhance its clarity and organization:

Structure: Organize the conclusion into clear sections: Summary of Findings, Limitations, Future Directions, and Final Remarks.

Redundancy: Remove repetitive content, such as multiple mentions of the "Leontief matrix" and acknowledgment of limitations.

Specify Key Findings: Highlight one or two main insights from the study.

Language Clarity: Simplify certain sentences for better clarity.

Citation Consistency: Ensure uniform citation formatting.

Broad Implications: Directly link the study's results to real-world applications, like policy shifts or investment strategies.

Implementing these suggestions will offer a clearer and more concise recap of the study's findings and its broader significance.

Reviewer 3 Report

Comments and Suggestions for Authors

My criterion for acceptance is that an article have results that are interesting. Thus being careful work and well expressed is not enough. This article in my view does not meet my criterion. I would not tell my classes about this study.

I encourage the authors to look for policy relevant results from their research, and to focus on those to hook the reader and inform policy makers.